# Effects of SGLT2 Inhibitors on Sleep Apnea Parameters and Cheyne–Stokes Respiration in Patients with Acute Decompensated Heart Failure: A Prospective Cohort Study

**DOI:** 10.3390/biomedicines13061474

**Published:** 2025-06-14

**Authors:** Petar Kalaydzhiev, Tsvetelina Velikova, Yanitsa Davidkova, Gergana Voynova, Angelina Borizanova, Natalia Spasova, Neli Georgieva, Radostina Ilieva, Elena Kinova, Assen Goudev

**Affiliations:** 1Department of Emergency Medicine, Medical University—Sofia, 1000 Sofia, Bulgaria; borizanowa@abv.bg (A.B.); spasova.natalia@gmail.com (N.S.); radostilieva@yahoo.com (R.I.); kinova.e@abv.bg (E.K.); agoudev@abv.bg (A.G.); 2Cardiology Department, University Hospital “Tsaritsa Yoanna—ISUL”, 1000 Sofia, Bulgaria; gerivoinova@gmail.com (G.V.); ngeorgieva87@gmail.com (N.G.); 3Medical Faculty, Sofia University St. Kliment Ohridski, 1407 Sofia, Bulgaria; tsvelikova@medfac.mu-sofia.bg; 4Department of Clinical Hematology, Specialized Hospital of Active Treatment of Hematological Diseases, 1000 Sofia, Bulgaria; yani.tihomirova@gmail.com

**Keywords:** SGLT2 inhibitors, acute decompensated heart failure, Cheyne–Stokes respiration, sleep apnea, right ventricular function

## Abstract

**Background:** Sleep-disordered breathing (SDB), particularly Cheyne–Stokes respiration (CSR), is highly prevalent among patients hospitalized with acute decompensated heart failure (ADHF) and is associated with worse clinical outcomes. Sodium-glucose cotransporter-2 inhibitors (SGLT2i) have demonstrated cardiorenal benefits in heart failure, but their effects on nocturnal respiratory parameters remain underexplored. **Objectives:** This study aims to evaluate the impact of SGLT2i therapy on key respiratory and cardiac indices including CSR burden, oxygenation, and right heart function in patients with ADHF and reduced left ventricular ejection fraction. **Methods:** In this single-center prospective cohort study, 60 patients with ADHF, LVEF < 40%, and a baseline apnea–hypopnea index (AHI) > 5 were assessed before and three months after the initiation of SGLT2i therapy. Sleep respiratory parameters were measured using home polygraphy (ApneaLink^TM^), while cardiac and renal indices were evaluated by echocardiography, NT-proBNP, and the estimated glomerular filtration rate (eGFR). Structural and functional echocardiographic changes were analyzed both at baseline and following the 3-month treatment period. Patient-reported outcomes were assessed using the Epworth Sleepiness Scale (ESS) and Kansas City Cardiomyopathy Questionnaire (KCCQ). **Results:** After 3 months of SGLT2i therapy, significant improvements were observed in daytime sleepiness (ESS: −2.68 points; *p* < 0.001), CSR index (−5.63 events/h; *p* < 0.001), AHI (−3.07 events/h; *p* < 0.001), ODI (−6.11 events/h; *p* < 0.001), and mean nocturnal SpO_2_ (+1.95%; *p* < 0.001). KCCQ scores increased by 9.16 points (*p* < 0.001), indicating improved quality of life. Cardiac assessments revealed reductions in NT-proBNP (−329.6 pg/mL; *p* < 0.001) and E/e′ ratio (−1.08; *p* < 0.001), with no significant change in LVEF or chamber dimensions. Right ventricular function improved, as evidenced by the increased TAPSE/sPAP ratio (+0.018; *p* < 0.001). Renal function remained stable, with a non-significant upward trend in eGFR. **Conclusions:** This exploratory study suggests that SGLT2 inhibitors may be associated with the attenuation of Cheyne–Stokes respiration and an improvement in right heart function in patients with ADHF, warranting further investigation in controlled trials. These findings highlight the potential of SGLT2is to address overlapping cardio-respiratory dysfunction in this high-risk population.

## 1. Introduction

ADHF frequently coexists with SDB, notably sleep apnea and Cheyne–Stokes respiration, both of which exacerbate cardiovascular strain and diminish patient quality of life. Sleep apnea in patients with ADHF contributes to intermittent nocturnal hypoxemia, enhanced sympathetic nervous system activity, and ventilatory instability, accelerating the progression of heart failure. CSR, characterized by cyclic fluctuations between hyperpnea and apnea, is a particularly common form of respiratory instability in this population and has been associated with poor prognosis, increased arrhythmogenic potential, and impaired sleep architecture [1,2].

A critical link between heart failure and sleep apnea is the pathophysiological impact of fluid redistribution. During recumbency, intravascular volume shifts rostrally toward the thoracic and peripharyngeal regions, predisposing patients to airway narrowing and altered respiratory mechanics [3]. This overnight rostral fluid shift plays a major role in the pathogenesis of sleep apnea in heart failure and has been identified as a potential therapeutic target [4,5]. The resulting pulmonary congestion and stimulation of vagal afferents contribute to periodic breathing and CSR, a phenomenon amplified by heightened chemoreflex sensitivity and autonomic dysregulation [6].

In recent years, SGLT2is have emerged as a class of cardioprotective agents with effects that extend far beyond glycemic control. In heart failure, SGLT2is promote natriuresis and osmotic diuresis, leading to reduced plasma volume, improved hemodynamic status, and attenuation of left ventricular filling pressures [7,8]. These effects may favorably influence SDB by limiting pulmonary congestion and mitigating fluid shifts during sleep [9,10,11]. Furthermore, SGLT2is have demonstrated the ability to reduce sympathetic overactivation, modulate chemoreceptor responsiveness, and potentially stabilize respiratory control networks involved in the generation of CSR [12,13,14]. Their emerging role in fluid management and nocturnal respiratory stabilization has positioned SGLT2is as promising candidates for targeting both cardiac and respiratory dysfunction in patients with ADHF [15].

Considering these factors, the present investigation was undertaken to elucidate the impact of SGLT2i therapy on nocturnal respiratory dynamics in patients admitted with ADHF and impaired left ventricular systolic function. Over the three-month treatment period, we systematically assessed changes in the incidence and severity of CSR, mean nocturnal oxygen saturation, and the overall burden of sleep apnea. These evaluations were accompanied by assessments of daytime somnolence and health-related quality of life, providing a comprehensive overview of the clinical impact of SGLT2 inhibition in this high-risk population.

## 2. Materials and Methods

This single-center, prospective cohort study was conducted at the Cardiology Department of University Hospital “Tsaritsa Yoanna—ISUL” in Sofia, Bulgaria, between January 2021 and December 2023. The study protocol was approved by the Ethics Committee of the Medical University of Sofia (Protocol 2145 from 27 April 2021), and all the patients provided written informed consent in accordance with the Declaration of Helsinki.

A total of 195 consecutively admitted patients with ADHF and LVEF < 40% were screened for eligibility. The inclusion criteria comprised NT-proBNP > 900 pg/mL and AHI > 5, as measured by the ApneaLink^TM^ system during hospitalization. Patients with NYHA class IV symptoms, end-stage renal failure, severe respiratory insufficiency, or COPD were excluded. Of the screened population, 73 patients met all the eligibility criteria and were enrolled in the study. Among the remaining 122 patients, 81 did not meet the inclusion criteria, while 14 were excluded due to comorbidities or unstable clinical condition. Additionally, 27 patients declined participation. After enrollment, 13 patients did not complete follow-up: 7 were lost to follow-up, 4 withdrew consent, and 2 died before the 3-month assessment. Thus, a total of 60 patients were included in the final analysis. Figure 1 illustrates the study screening and enrollment process.

SDB was assessed using ApneaLink^TM^ (ResMed^®^ San Diego, CA, USA), a validated home respiratory polygraphy device that records nasal airflow, thoracic movements, pulse rate, and peripheral oxygen saturation (SpO_2_). ApneaLink^TM^ has demonstrated high diagnostic agreement with polysomnography and is widely used in real-world sleep apnea screening [16]. Derived parameters included the apnea-hypopnea index (AHI), oxygen desaturation index (ODI), presence of CSR, and average and minimum nocturnal SpO_2_.

Echocardiographic evaluation and NT-proBNP measurements were performed at baseline and at the 3-month follow-up. Functional outcomes were assessed using the Epworth Sleepiness Scale (ESS) and the Kansas City Cardiomyopathy Questionnaire (KCCQ). The ESS is a standardized and validated instrument used to quantify subjective daytime sleepiness, with strong internal consistency and test–retest reliability [17]. The KCCQ is a disease-specific health status questionnaire that evaluates symptoms, physical limitations, quality of life, and social functioning in patients with heart failure. It has shown high sensitivity to clinical change and is considered a robust tool for measuring patient-reported outcomes in this population [18].

Descriptive statistics were used for baseline characterization. Continuous variables were expressed as mean ± standard deviation (SD). Normality was tested using the Kolmogorov–Smirnov test. Within-group comparisons between baseline and follow-up values were conducted using the paired-samples t-test. Given the exploratory design and limited sample size, multivariable regression analyses were not conducted in order to avoid overfitting and unstable estimates. Paired comparisons were prioritized to evaluate within-subject changes over time. A *p*-value < 0.05 was considered statistically significant. All analyses were performed using IBM SPSS Statistics, version 24.

## 3. Results

### 3.1. Baseline Characteristics

The cohort comprised predominantly male patients in their seventh decade of life, with non-ischemic etiology, atrial fibrillation, and type 2 diabetes mellitus each affecting a substantial proportion—hallmarks of the complex comorbidity profile in ADHF. Guideline-directed therapies were broadly implemented: most patients received β-blockers, ACEI/ARB, ARNi, MRAs, and loop diuretics. SGLT2i initiation—either dapagliflozin or empagliflozin—occurred in roughly half of the population, with dosing tailored to individual clinical needs. Among the patients with type 2 diabetes (n = 34), 21 (61.8%) were treated with metformin, 8 (23.5%) with insulin, 6 (17.6%) with DPP-4 inhibitors, and 2 (5.9%) with sulfonylureas. No patients received GLP-1 receptor agonists or thiazolidinediones. Throughout the 3-month follow-up, no symptomatic hypoglycemia, hyperglycemic crises, or cases of ketoacidosis were reported. These baseline features mirror those of contemporary heart failure trials and real-world registries, supporting the external validity of our findings (Table 1).

### 3.2. Sleep-Related Respiratory Outcomes

Following three months of SGLT2i therapy, the patients demonstrated a significant attenuation of SDB, as detailed in Table 2. Subjective daytime sleepiness, quantified by the ESS, decreased appreciably, coinciding with a reduction in the frequency of apnea–hypopnea events per hour (AHI). The measures of nocturnal oxygenation improved, with a lower oxygen desaturation index and higher mean overnight SpO_2_, while the prevalence and duration of the Cheyne–Stokes respiration cycles declined. These respiratory improvements were paralleled by enhanced health-related quality of life, reflected in a marked increase in KCCQ overall scores, indicating relief of heart failure-related symptoms and functional impairment (Table 2).

### 3.3. Cardiac and Functional Parameters

Concomitant cardiac assessments revealed neurohormonal and hemodynamic stabilization without a compromise of ventricular systolic performance. NT-proBNP concentrations fell, consistent with reduced ventricular wall stress, and the E/e′ ratio improved, indicating lower left ventricular filling pressures. The left ventricular ejection fraction remained unchanged, and the chamber dimensions—left ventricular end-diastolic volume and left atrial volume index—exhibited minimal variation, suggesting that structural reverse remodeling was not yet pronounced. The right heart evaluation showed a modest decrease in RVOT diameter with a stable right atrial area, whereas TAPSE increased, denoting improved right ventricular contractile function; the relationship between TAPSE and systolic pulmonary artery pressure is illustrated in Figure 2. Renal function, as assessed by both eGFR and serum creatinine, remained preserved throughout the study period, with a slight but non-significant decrease in serum creatinine. Additionally, a significant reduction in body mass index (BMI) was observed. These favorable changes likely reflect the effective decongestive therapy achieved during hospitalization and follow-up, contributing to improved volume status and metabolic profile.

All the findings are detailed in Table 3.

## 4. Discussion

The interplay between ADHF and SDB, particularly CSR, is rooted in a complex pathophysiological framework involving chronic hyperventilation, prolonged circulatory delay, and enhanced chemoreceptor sensitivity [19]. Our study supports this model by identifying a high prevalence of CSR in a cohort of ADHF patients with reduced LVEF. One of the central mechanisms likely underpinning this relationship is the rostral fluid shift during recumbency, which facilitates pulmonary congestion, upper airway narrowing, and cyclical apnea–hyperpnea patterns that typify CSR [20,21]. These dynamics are not merely mechanical but are further compounded by autonomic instability and arrhythmogenic substrates. In this context, autonomic dysregulation and coexisting AF act synergistically to exacerbate ventilatory instability, creating a self-reinforcing loop of nocturnal hypoxemia, sympathetic activation, and progressive cardiac decompensation [22]. These interrelated processes reinforce the emerging concept that CSR is not merely a by-product of heart failure but represents a modifiable pathophysiological target with both prognostic and therapeutic implications. Recent data further support this interplay, highlighting the potential of both SGLT2 inhibitors and GLP-1 receptor agonists to mitigate cardiovascular risk and reduce OSA severity through pleiotropic, multi-organ effects [23].

Although autonomic modulation and fluid redistribution are biologically plausible mechanisms through which SGLT2is may attenuate CSR, our study did not include direct assessments of sympathetic activity or fluid volume status. Thus, the mechanistic basis of these observations remains to be elucidated in future studies designed to evaluate these physiological domains more directly.

Simultaneously, the demographic and clinical profile of our cohort provides essential context for the interpretation of our findings. It reflects a representative phenotype of patients with ADHF and reduced LVEF, aligning with those observed in pivotal heart failure trials and large observational registries. The predominance of elderly males, the high prevalence of AF and T2DM, and a majority non-ischemic etiology enhance the generalizability of our findings [24,25,26]. From a therapeutic standpoint, all the patients were initiated on SGLT2i therapy upon discharge, and dosing strategies were tailored to clinical status and comorbidity profile. Lower doses were predominantly prescribed to T2DM patients already receiving combination antihyperglycemic agents such as metformin or DPP-4 inhibitors, and to those with advanced age or a higher risk for intravascular volume depletion. This approach is consistent with real-world prescribing behaviors and reflects a precision-based strategy in the implementation of guideline-directed medical therapy for HFrEF [27,28,29].

Consistent with this optimized pharmacological regimen, the three-month follow-up revealed substantial improvements in multiple parameters of sleep-disordered breathing. Specifically, the ESS scores decreased significantly, indicating the amelioration of daytime somnolence, while reductions in both AHI and ODI underscored improvements in nocturnal respiratory stability and oxygenation. These changes were accompanied by a meaningful rise in mean overnight SpO_2_, albeit without a statistically significant shift in nadir saturation. The improvements observed across these objective and subjective domains underscore the potential role of SGLT2is in modulating nocturnal breathing patterns. Of particular relevance is that the CSR index declined by more than five events per hour, consistent with emerging data that suggest SGLT2is mitigate respiratory instability through hemodynamic modulation and the attenuation of sympathetic tone [30,31]. The clinical importance of these improvements is further substantiated by a marked increase in KCCQ scores, indicating enhanced patient-reported symptom burden and quality of life. These findings are in agreement with results from the DAHOS study and the meta-analysis by Polecka et al., both of which demonstrated the consistent benefits of SGLT2is for sleep quality, ventilatory parameters, and functional outcomes in HF patients with coexisting SDB [31].

The cardiac biomarker and echocardiographic data further reinforced the systemic benefits of SGLT2i in this high-risk cohort. The significant reduction in NT-proBNP, coupled with a decrease in E/e′, suggests effective LV unloading and improved diastolic filling pressures. These effects are mechanistically in line with the known natriuretic and vasodilatory actions of SGLT2is and mirror the findings of Correale et al., who observed similar improvements in patients undergoing combination therapy with SGLT2is and ARNis [32]. Similarly, Jaffuel et al. [33] investigated ARNi-based therapy optimization in patients with reduced systolic function and re-evaluated sleep-disordered breathing parameters after three months. They also reported significant reductions in central apneas and Cheyne–Stokes respiration, along with improved nocturnal oxygen saturation. These findings reinforce the importance of optimizing pharmacological treatment according to current guidelines before initiating positive airway pressure therapy for sleep apnea at home. While LVEF remained stable over the observation period, modest changes in LVEDV and LAVi suggest the possibility of early reverse remodeling, which may become more evident with longer treatment duration. This is consistent with prior studies indicating that functional improvements often precede structural adaptation during SGLT2i therapy [34]. In addition, beyond their cardiorenal effects, SGLT2 inhibitors may influence chemoreflex sensitivity and autonomic tone, potentially modulating central respiratory drive. Their emerging antiarrhythmic and sympathoinhibitory properties may further stabilize ventilatory patterns, particularly in patients with underlying autonomic imbalance or atrial fibrillation [35].

Complementing the left-sided findings, additional insights were obtained from the assessment of right ventricular function. An increase in TAPSE, along with a reduction in RVOT diameter and stable RA area, suggests improved RV contractile performance. Furthermore, the observed rise in the TAPSE/sPAP ratio reflects enhanced RV–PA coupling, an integrated echocardiographic indicator of right ventricular adaptation to afterload. Notably, while this improvement occurred in parallel with attenuation of CSR burden, the relationship should be interpreted as associative rather than causal. This parameter has gained recognition as a strong independent prognostic marker in patients with HF, particularly in the presence of pulmonary hypertension and SDB [36,37]. Given the mechanistic link between RV dysfunction and CSR, our findings further support the notion that improvement in RV–PA coupling may mediate some of the observed respiratory benefits [38].

Renal function was also closely monitored, given the well-established nephroprotective effects of SGLT2is. eGFR remained stable throughout the observation period, with a non-significant trend toward improvement. This trajectory mirrors the expected biphasic renal response to SGLT2i therapy, wherein an early and transient rise in serum creatinine attributed to afferent arteriolar vasoconstriction is followed by the functional stabilization or recovery of glomerular filtration [39]. Although SGLT2 inhibitors are associated with an initial decline in eGFR during the first weeks of therapy, previous studies have demonstrated a subsequent recovery phase. In our cohort, the modest improvement in eGFR observed after 3 months may reflect both this physiological rebound and the effects of optimized heart failure therapy during the post-decompensation period. Importantly, no adverse renal events were observed, which further supports the safety and tolerability of SGLT2is in this clinically fragile population.

## 5. Limitations

While this study contributes valuable insights into the intersection of heart failure and sleep-disordered breathing, a few limitations should be acknowledged. Being conducted at a single center, the findings may not fully capture the diversity of patient populations or practice patterns in other settings. The sample size, though adequate for evaluating overall treatment effects, was not large enough to explore subgroup differences or interactions, such as those based on heart failure etiology or the initial severity of CSR. Additionally, the use of portable respiratory polygraphy, while practical and validated, offers a limited view of sleep architecture and does not distinguish central from obstructive events with the granularity of full polysomnography. We did not record or control for patients’ sleeping position, which may have influenced the detection and severity of sleep-disordered breathing events. We did not assess acid–base status or systematically exclude noncardiac causes of CSR, such as uremia, CNS pathology, or medication-induced effects. The three-month follow-up period, although sufficient to detect early physiological responses, may not reflect the full trajectory of long-term cardiac or renal adaptation. Moreover, the lack of multivariate modeling represents a limitation, as it precludes adjustment for potential confounders such as age, baseline disease severity, and comorbidities. Our findings generate hypothesis-supporting evidence that SGLT2 inhibitors may influence respiratory and cardiac parameters in patients with ADHF. However, due to the limitations of the study design, these observations should be interpreted with caution.

## 6. Conclusions

This prospective cohort study suggests that SGLT2i therapy in patients with ADHF and reduced LVEF may be associated with early improvements in SDB, RV function, and markers of decongestion. The favorable trends observed in both the respiratory parameters and patient-reported outcomes support the hypothesis that SGLT2is exert pleiotropic benefits beyond glycemic control. While promising, these findings should be interpreted cautiously due to the limited sample size, lack of a comparator arm, and short follow-up. Future studies with larger cohorts are needed to enable more robust statistical control.

## Figures and Tables

**Figure 1 biomedicines-13-01474-f001:**
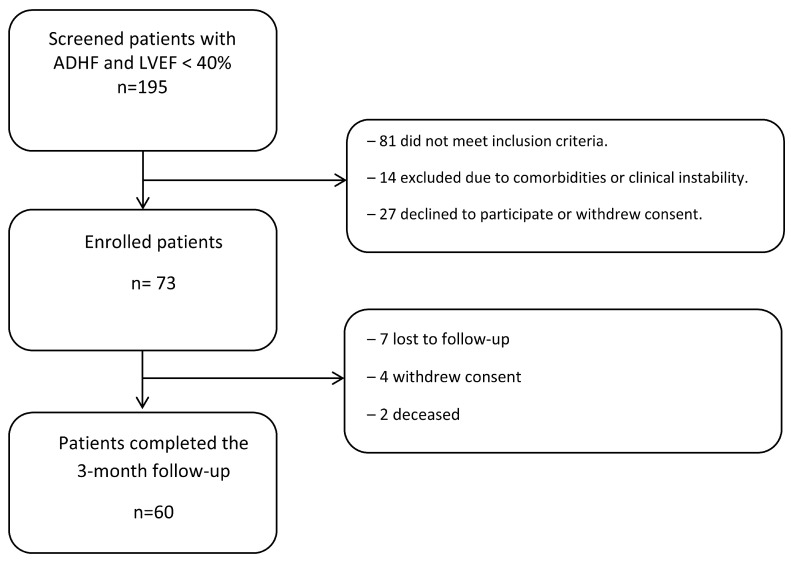
Study flow diagram. Of 195 ADHF patients (LVEF < 40%) screened, 73 were enrolled; 13 were excluded during the 3-month follow-up, leaving 60 for final analysis. ADHF—acute decompensated heart failure; LVEF—left ventricular ejection fraction.

**Figure 2 biomedicines-13-01474-f002:**
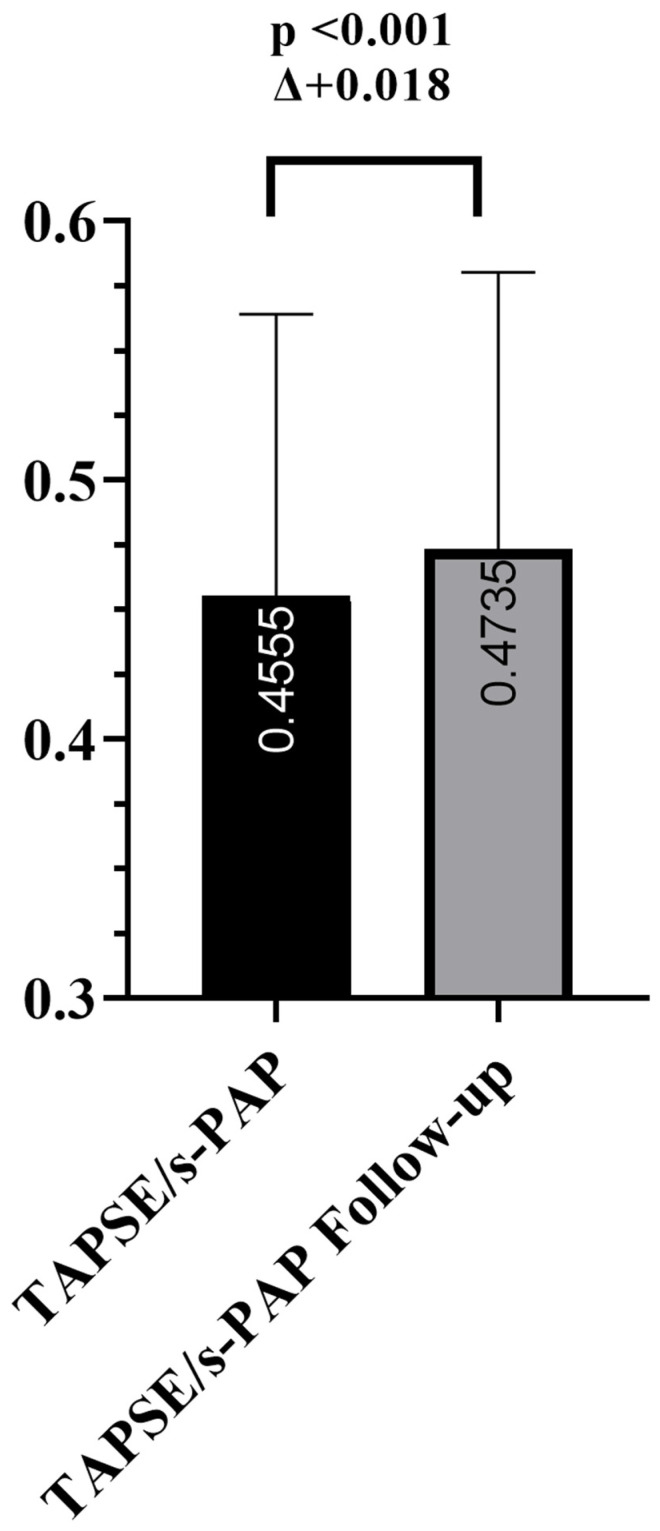
Change in TAPSE/sPAP ratio following 3 months of SGLT2i therapy. Bar chart illustrating the increase in the TAPSE/sPAP ratio—an integrated echocardiographic marker of right ventricular–pulmonary arterial coupling—between baseline and 3-month follow-up in patients with acute decompensated heart failure. The observed improvement reflects enhanced right ventricular systolic function relative to pulmonary arterial pressure.

**Table 1 biomedicines-13-01474-t001:** Baseline characteristics of the study cohort (N = 60).

Characteristic	Value, n (%)
**Age, years**	69.3 ± 9.4 (45–86)
**Sex, n (%)**	
Female	19 (31.7)
Male	41 (68.3)
**Non-ischemic etiology, n (%)**	38 (63.3)
**Atrial fibrillation, n (%)**	43 (71.7)
**Diabetes mellitus type 2, n (%)**	34 (56.7)
**Medications, n (%)**	
Beta-blockers	54 (90.0)
ACEI/ARB	44 (73.3)
ARNi	33 (55.0)
MRA	26 (43.3)
Loop diuretics	52 (86.7)
**Antidiabetic therapy among patients with DM2**	
Dapagliflozin 5 mg	10 (16.7)
Dapagliflozin 10 mg	26 (43.3)
Empagliflozin 5 mg	3 (5.0)
Empagliflozin 10 mg	21 (35.0)
Metformin	21 (61.8%)
DPP-4 inhibitors	6 (17.6%)
Insulin (basal and/or bolus)	8 (23.5%)
Sulfonylureas	2 (5.9%)

Continuous data are presented as mean ± SD (range); categorical data as n (%). Abbreviations: ACEI—angiotensin-converting enzyme inhibitor; ARB—angiotensin II receptor blocker; ARNi—angiotensin receptor–neprilysin inhibitor; MRA—mineralocorticoid receptor antagonist; DM2—type 2 diabetes mellitus; DPP-4 inhibitors—dipeptidyl peptidase-4 inhibitors; SD—standard deviation.

**Table 2 biomedicines-13-01474-t002:** Sleep-related and patient-reported outcomes at baseline and 3-month follow-up.

Parameter	Baseline	Follow-Up	Δ (Follow-Up − Baseline)	*p*-Value
ESS	10.72 ± 1.61	8.04 ± 2.27	−2.68	<0.001
KCCQ overall score	62.87 ± 5.73	72.03 ± 6.99	+9.16	<0.001
CSR index (events/h)	32.63 ± 11.05	27.01 ± 9.61	−5.63	<0.001
AHI (events/h)	21.18 ± 4.82	18.11 ± 4.64	−3.07	<0.001
ODI (events/h)	24.35 ± 6.66	18.24 ± 5.83	−6.11	<0.001
Mean nocturnal SpO_2_ (%)	89.98 ± 2.85	91.93 ± 2.58	+1.95	<0.001
Lowest nocturnal SpO_2_ (%)	80.91 ± 5.55	81.69 ± 5.26	+0.78	0.206

Data are mean ± SD; Δ indicates absolute change from baseline to follow-up. ESS—Epworth Sleepiness Scale; KCCQ—Kansas City Cardiomyopathy Questionnaire; CSR—Cheyne–Stokes respiration; AHI—apnea–hypopnea index; ODI—oxygen desaturation index; SpO_2_—peripheral oxygen saturation.

**Table 3 biomedicines-13-01474-t003:** Cardiac, renal, and echocardiographic parameters at baseline and 3-month follow-up.

Parameter	Baseline	Follow-Up	Δ (Follow-Up − Baseline)	*p*-Value
NT-proBNP (pg/mL)	1780.9 ± 882.8	1451.3 ± 923.4	−329.64	<0.001
E/e′	14.55 ± 2.76	13.48 ± 2.65	−1.08	<0.001
LVEF (%)	35.06 ± 4.65	35.16 ± 4.85	+0.10	0.490
LVEDV (mL)	191.76 ± 19.20	189.78 ± 17.97	−1.98	0.094
LAVi (mL/m^2^)	45.58 ± 4.20	45.47 ± 4.24	−0.11	0.239
eGFR (mL/min/1.73 m^2^)	61.87 ± 9.42	62.37 ± 9.56	+0.50	0.210
Serum creatinine, µmol/L	108 ± 9.1	103 ± 8.2	−5	0.081
BMI, kg/m^2^	29.2 ± 2.6	27.8 ± 2.5	−1.4	0.011
RVOT diameter (mm)	39.44 ± 4.18	38.53 ± 3.91	−0.92	0.001
RA area (cm^2^)	21.93 ± 3.15	21.99 ± 3.04	+0.06	0.728
TAPSE/s-PAP (mm/mmHg)	0.455 ± 0.108	0.474 ± 0.107	+0.018	<0.001

Data are mean ± SD; Δ indicates absolute change from baseline to follow-up. NT-proBNP—N-terminal pro-B-type natriuretic peptide; E/e′—ratio of early mitral inflow velocity to mitral annular early diastolic velocity; LVEF—left ventricular ejection fraction; LVEDV—left ventricular end-diastolic volume; LAVi—left atrial volume index; eGFR—estimated glomerular filtration rate; RVOT—right ventricular outflow tract; RA—right atrium; TAPSE—tricuspid annular plane systolic excursion; s-PAP—systolic pulmonary artery pressure; BMI—body mass index.

## Data Availability

Data cannot be shared for ethical/privacy reasons. The data underlying this article cannot be shared publicly due to ethical reasons. The data contain sensitive in-formation and are associated with questionnaires completed by patients. The data will be shared upon reasonable request to the corresponding author, after any sensitive information has been removed.

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
