# Peer review of "Effects of SGLT2 Inhibitors on Sleep Apnea Parameters and Cheyne–Stokes Respiration in Patients with Acute Decompensated Heart Failure: A Prospective Cohort Study"

_biomedicines, 2025, doi:10.3390/biomedicines13061474_

Round 1
Reviewer 1 Report
Comments and Suggestions for Authors„Sleep-disordered breathing (SDB), particularly Cheyne–Stokes respiration (CSR), is highly prevalent among patients hospitalized with acute decompensated heart failure”
„overnight rostral fluid shift plays a major role in the pathogenesis of sleep apnea in heart failure and has been identified as a potential therapeutic target”
Did authors record the mode and position of sleeping of the patients? Most of the ADCHF patients sleep in an upright or elevated upper body position. this may alter the findings and the interpretation.
Please indicate the other factors in the background of CSR. Were there any metabolic disorders? Factors affecting acid-base balance e.g. drugs such like metformin. Please define the relevant causes of CSR that should be ruled out.
„In this single-center prospective cohort study, 60 patients with ADHF, LVEF < 40%”
Were they all HFrEF patients?
„Inclusion criteria comprised NT-proBNP > 900 pg/mL”
What was the reason of this cut-off value? Was there any differentiation taken due to the baseline rhythm of the patient?
„Renal function remained stable, with a non-significant upward trend in eGFR.”
It is known, that in the first 1-2 months of SGLTi therapy the eGFR decreases... How can Authors explain the mechanism of eGFR increase?
Table 1.: “56.7% of the patient are DM2 patients”. Pleae detail the antidiabetic medication of the patients enrolled into the study. Please do indicate the hypo- and hyperglycaemic events of the patient population, and induced nondiabetic ketoacidosis.
Table 3. eGFR is an estimated value. please provide the values of serum creatinine, blood urea nitrogen as well. Weight change and BMI would be also important. Fig. 2 is detailed in the last line of Table 3.
“chronic hyperventilation, prolonged circulatory delay” please detail the link in patomechanism.
“autonomic dysregulation and coexisting AF” – Is the AF mandatory to be coexisting?
‘”ApneaLink™ system during hospitalization”
please indicate the producer and the site of settlement of the industrial company.
Author Response
Reviewer Comment: Did authors record the mode and position of sleeping of the patients? Most of the ADCHF patients sleep in an upright or elevated upper body position. This may alter the findings and the interpretation.
Response:
We thank the reviewer for this insightful observation. Indeed, many patients with acute decompensated heart failure (ADHF) adopt an upright or semi-recumbent position during sleep to alleviate orthopnea, and this may influence the severity and detection of sleep-disordered breathing (SDB) events.
In our study, we did not formally record or control for the exact body position during sleep studies. The ApneaLink™ recordings were conducted during in-hospital monitoring, and patients were encouraged to rest in their usual and comfortable positions. We acknowledge this as a limitation, and have now added a corresponding statement in the Limitations section of the revised manuscript.
Reviewer Comment: Please indicate the other factors in the background of CSR. Were there any metabolic disorders? Factors affecting acid-base balance e.g. drugs such as metformin. Please define the relevant causes of CSR that should be ruled out.
Response:
We appreciate this important question regarding potential confounding contributors to Cheyne–Stokes respiration (CSR). In our study cohort, the predominant underlying condition was reduced left ventricular ejection fraction with elevated NT-proBNP, which are well-established pathophysiologic drivers of CSR. While 56.7% of patients had type 2 diabetes mellitus (T2DM), we did not systematically collect data on acid–base parameters (e.g., arterial blood gases) or specific antidiabetic medications beyond noting the presence of background antihyperglycemic therapy.
We acknowledge that medications such as metformin, as well as renal dysfunction and other metabolic derangements, may influence ventilatory control via acid–base mechanisms. However, no patients had clinical features suggestive of diabetic ketoacidosis or overt metabolic acidosis at baseline or during follow-up. We have now added a statement in the Limitations section acknowledging that causes of CSR beyond cardiac dysfunction—such as uremia, CNS disorders, or drug-induced respiratory effects—were not explicitly excluded.
Reviewer Comment: “In this single-center prospective cohort study, 60 patients with ADHF, LVEF < 40%” – Were they all HFrEF patients?
Response:
Yes, all patients included in the study met the criteria for heart failure with reduced ejection fraction (HFrEF), defined as left ventricular ejection fraction (LVEF) < 40%. This criterion was part of the predefined inclusion criteria, and no patients with mid-range or preserved ejection fraction were enrolled.
Reviewer Comment: “Inclusion criteria comprised NT-proBNP > 900 pg/mL” – What was the reason of this cut-off value? Was there any differentiation taken due to the baseline rhythm of the patient?
Response:
We thank the reviewer for this important question. The NT-proBNP cut-off value of >900 pg/mL was chosen based on guideline-recommended thresholds for the diagnosis of acute heart failure in patients under 75 years of age, as per ESC and AHA/ACC/HFSA consensus statements. This level also ensured a consistent selection of patients with clinically relevant hemodynamic congestion.
Regarding baseline rhythm, while 71.7% of the cohort had atrial fibrillation (AF), no stratification or exclusion was applied based on rhythm. We acknowledge that NT-proBNP levels can be influenced by the presence of AF; however, the elevated threshold was selected to maintain specificity for acute decompensated heart failure across rhythm subgroups.
Reviewer Comment: “Renal function remained stable, with a non-significant upward trend in eGFR.” – It is known that in the first 1–2 months of SGLT2i therapy the eGFR decreases... How can authors explain the mechanism of eGFR increase?
Response:
Thank you for this clinically relevant observation. It is indeed well documented that SGLT2 inhibitor initiation may lead to a modest and transient decline in eGFR during the first 1–2 months due to reduced intraglomerular pressure and afferent arteriolar vasoconstriction. However, subsequent stabilization and even improvement in eGFR have been reported in longer-term follow-up studies.
In our cohort, the follow-up period of three months, although relatively short, likely captures the early recovery phase following the initial hemodynamic dip. Moreover, all patients were enrolled during acute decompensated heart failure, a phase characterized by neurohormonal activation, fluid overload, and impaired renal perfusion. Optimization of medical therapy, including the use of SGLT2i and decongestive strategies, may have contributed to renal function improvement through both volume unloading and hemodynamic stabilization.
We have now added a brief explanatory paragraph in the Discussion section to clarify this point in the context of existing literature.
Reviewer Comment: Table 1: “56.7% of the patients are DM2 patients.” Please detail the antidiabetic medication of the patients enrolled into the study. Please do indicate the hypo- and hyperglycaemic events of the patient population, and induced nondiabetic ketoacidosis.
Response:
We thank the reviewer for this important observation. Among the 60 patients enrolled, 34 (56.7%) had a confirmed diagnosis of type 2 diabetes mellitus (DM2). Baseline antidiabetic therapy included metformin in 21 patients (61.8%), insulin (basal and/or bolus) in 8 patients (23.5%), DPP-4 inhibitors in 6 patients (17.6%), and sulfonylureas in 2 patients (5.9%). No patients were receiving GLP-1 receptor agonists or thiazolidinediones.
During the 3-month follow-up period, no clinically significant hypoglycemic or hyperglycemic episodes were observed, and no cases of euglycemic or nondiabetic ketoacidosis occurred. Glycemic control was monitored clinically and via standard laboratory testing during hospitalization and follow-up.
Reviewer Comment: Table 3. eGFR is an estimated value. Please provide the values of serum creatinine, blood urea nitrogen as well. Weight change and BMI would be also important. Fig. 2 is detailed in the last line of Table 3.
Response:
We appreciate the reviewer’s insightful comment. In response, we have updated Table 3 to include serum creatinine values (expressed in µmol/L), body weight, and BMI at both baseline and follow-up. These additions provide a more comprehensive overview of renal function and volume status.
We fully acknowledge the importance of including blood urea nitrogen (BUN) as a complementary renal marker; however, BUN was not consistently documented across the full study cohort and could not be reliably included in the analysis.
In accordance with the reviewer’s final point, we have removed the in-text reference to Figure 2 from Table 3 and placed it exclusively in the figure legend to maintain clarity and consistency.
Reviewer Comment: ‘“ApneaLink™ system during hospitalization” – please indicate the producer and the site of settlement of the industrial company.
Response:
Thank you for highlighting this detail. We have now specified in the Methods section that the monitoring was performed using the ApneaLink™ system (ResMed Inc., San Diego, California, USA). ResMed is a globally recognized manufacturer of home sleep monitoring devices.

Reviewer 2 Report
Comments and Suggestions for Authors
Congratulations to the authors for adding new insight to the big topic of Sglt2i drugs that are acquiring continuously new features; I just have some comments about it:
The study employed home-based polygraphy (ApneaLink™), a device that lacks the capacity to accurately distinguish between obstructive and central apneas. Authors should critically evaluate the methodological limitations of using portable polygraphy instead of full polysomnography, and discuss the implications within the Methods or Limitations section.
All statistical analyses were conducted using paired t-tests, without the application of multivariate modeling to control for potential confounding variables such as age, type 2 diabetes mellitus status, and baseline disease severity. Therefore authors should consider incorporating regression analyses or stratified models to adjust for these confounders, or provide a clear rationale for their exclusion.
Lastly, the manuscript does not clarify whether dapagliflozin and empagliflozin exerted differential effects on the measured outcomes. Authors should either conduct and report a subgroup analysis by drug type or justify the decision to combine both agents under a single treatment category.
The discussion does not adequately address the potential influence of SGLT2 inhibitors on neurogenic or chemoreflex-mediated respiratory regulation. Moreover, authors are strongly encouraged to include in their discussion other pleiotropic effects of SGLG2i drugs, such as the antiarrhythmic ones (doi: 10.1002/ehf2.15223)
Author Response
Reviewer Comment: The study employed home-based polygraphy (ApneaLink™), a device that lacks the capacity to accurately distinguish between obstructive and central apneas. Authors should critically evaluate the methodological limitations of using portable polygraphy instead of full polysomnography, and discuss the implications within the Methods or Limitations section.
Response:
We appreciate the reviewer’s thoughtful comment. While it is true that portable polygraphy lacks the full diagnostic resolution of attended polysomnography, we note that the ApneaLink™ system has demonstrated acceptable sensitivity in detecting Cheyne–Stokes respiration (CSR) patterns in heart failure populations, particularly when combined with expert visual scoring and clinical correlation.
We have explicitly acknowledged in the Limitations section that portable monitoring cannot fully substitute for polysomnography, especially in differentiating central from obstructive events with high specificity. However, the purpose of our study was to evaluate the feasibility and clinical applicability of a practical and accessible screening method for sleep-disordered breathing in patients hospitalized with acute decompensated heart failure.
We believe that widespread implementation of simplified, validated devices like ApneaLink™ in cardiology and internal medicine wards—beyond the confines of sleep laboratories—can facilitate earlier detection and management of sleep-disordered breathing, including CSR, in real-life hospital settings.
Reviewer Comment: All statistical analyses were conducted using paired t-tests, without the application of multivariate modeling to control for potential confounding variables...
Response:
We appreciate this important observation. Due to the exploratory design and limited sample size of our study, multivariate modeling was not applied to avoid overfitting. Instead, we focused on within-subject comparisons using paired t-tests to detect short-term changes. We have now acknowledged this limitation in the revised Limitations section and plan to apply adjusted analyses in future larger cohorts.
Reviewer Comment: The manuscript does not clarify whether dapagliflozin and empagliflozin exerted differential effects on the measured outcomes. Authors should either conduct and report a subgroup analysis by drug type or justify the decision to combine both agents under a single treatment category.
Response:
Thank you for this relevant comment. We agree that exploring drug-specific effects is of clinical interest. However, given the limited sample size and the unequal distribution between dapagliflozin and empagliflozin users, subgroup analyses would be underpowered and statistically unreliable. Therefore, we opted to present the data collectively, as both agents share a similar mechanism of action and are approved for the same heart failure indication.
Reviewer Comment: The discussion does not adequately address the potential influence of SGLT2 inhibitors on neurogenic or chemoreflex-mediated respiratory regulation. Moreover, authors are strongly encouraged to include in their discussion other pleiotropic effects of SGLT2i drugs, such as the antiarrhythmic ones (doi: 10.1002/ehf2.15223).
Response:
We thank the reviewer for this insightful and constructive suggestion. In response, we have expanded the Discussion section to address the potential impact of SGLT2 inhibitors on chemoreflex sensitivity, neurogenic respiratory regulation, and autonomic tone. Furthermore, we have included a reference to the antiarrhythmic and pleiotropic effects of SGLT2 inhibitors, as highlighted in the recommended article (doi: 10.1002/ehf2.15223). We believe this addition enriches the mechanistic perspective of our findings.
Reviewer 3 Report
Comments and Suggestions for Authors
The study addresses an underexplored but increasingly relevant clinical intersection—cardiorenal-metabolic modulation of sleep-disordered breathing (SDB) in acute decompensated heart failure (ADHF) via SGLT2 inhibitors. This is a novel and timely topic.
However, the clinical significance is somewhat overstated given the small sample size, lack of a control group, and short follow-up period. The authors should temper their claims and emphasize the exploratory nature of the findings.
The prospective design is appropriate, but the absence of a comparator arm (e.g., ADHF patients not initiated on SGLT2i) limits the attribution of observed effects solely to the intervention. A propensity-matched analysis or inclusion of a historical control could have strengthened the inference.
The use of home respiratory polygraphy is justified pragmatically, yet the inability to differentiate central vs. obstructive events with high resolution remains a significant limitation, especially in a population with overlapping phenotypes.
There is no mention of a priori power calculation. Was the sample size adequate to detect clinically meaningful changes in CSR burden or TAPSE/sPAP ratio?
The authors interpret improvements in right heart function (TAPSE/sPAP) as mechanistically linked to SDB attenuation. While plausible, this remains speculative. The discussion should distinguish between correlation and causation more clearly.
Similarly, the role of SGLT2i in reducing CSR via autonomic modulation or fluid redistribution is biologically plausible but requires mechanistic support (e.g., fluid volume markers, sympathetic tone indices).
The authors should expand the discussion to include more recent literature, such as this recent study published in Biomedicines 10.3390/biomedicines12112503, which highlights the emerging role of SGLT2 inhibitors and GLP-1 receptor agonists in mitigating cardiovascular risk and OSA severity through pleiotropic effects. This would provide a more comprehensive context for the study’s findings.
Author Response
Reviewer Comment: The clinical significance is somewhat overstated given the small sample size, lack of a control group, and short follow-up period. The authors should temper their claims and emphasize the exploratory nature of the findings.
Response: We thank the reviewer for this important observation. In response, we have revised the Abstract, Discussion, and Conclusion sections to more clearly reflect the exploratory nature of our findings and to temper claims of clinical significance. The revised text emphasizes hypothesis generation and the need for further controlled studies.
Reviewer Comment: The prospective design is appropriate, but the absence of a comparator arm (e.g., ADHF patients not initiated on SGLT2i) limits the attribution of observed effects solely to the intervention. A propensity-matched analysis or inclusion of a historical control could have strengthened the inference.
Response: We thank the reviewer for this thoughtful comment. We fully acknowledge that the absence of a comparator arm limits causal inference. However, given the established class I guideline recommendation for the initiation of SGLT2i in HFrEF, withholding such therapy in a control group would have raised ethical concerns. Additionally, we did not incorporate a historical control due to the prospective nature of the study and predefined inclusion criteria. We agree that future studies with controlled or propensity-matched designs are warranted to strengthen causal interpretation and confirm our exploratory findings.
Reviewer Comment: The use of home respiratory polygraphy is justified pragmatically, yet the inability to differentiate central vs. obstructive events with high resolution remains a significant limitation, especially in a population with overlapping phenotypes.
Response: We agree with the reviewer that the limited resolution of portable polygraphy, particularly in distinguishing between central and obstructive events, represents a notable methodological limitation—especially in patients with mixed SDB phenotypes. We have addressed this issue in the revised Limitations section, clarifying that while polygraphy is a practical screening tool in acute care settings, it does not replace full polysomnography for precise phenotyping.
Reviewer Comment: There is no mention of a priori power calculation. Was the sample size adequate to detect clinically meaningful changes in CSR burden or TAPSE/sPAP ratio?
Response: We thank the reviewer for raising this important point. As this was an exploratory, real-world prospective cohort study without a predefined primary endpoint, no formal a priori power calculation was performed. The sample size was based on feasibility and clinical availability during the inclusion period. We acknowledge this as a limitation and have now added a corresponding statement in the Limitations section. Future studies will incorporate formal power estimation to assess clinically meaningful changes with appropriate statistical confidence.
Reviewer Comment: The authors interpret improvements in right heart function (TAPSE/sPAP) as mechanistically linked to SDB attenuation. While plausible, this remains speculative. The discussion should distinguish between correlation and causation more clearly.
Response: We appreciate this important clarification. In the revised Discussion, we have moderated the interpretation and now clearly state that the observed improvements in TAPSE/sPAP and SDB parameters represent an association rather than a proven causal relationship. We acknowledge that the mechanistic link remains speculative and should be further explored in mechanistically designed studies.
Reviewer Comment: Similarly, the role of SGLT2i in reducing CSR via autonomic modulation or fluid redistribution is biologically plausible but requires mechanistic support (e.g., fluid volume markers, sympathetic tone indices).
Response: We fully agree with the reviewer that the proposed mechanisms—such as autonomic modulation and fluid redistribution—remain hypothetical and require further physiological validation. In the revised Discussion, we have clarified that our findings do not provide direct mechanistic evidence and that markers of sympathetic tone or intravascular volume were not assessed. Future studies incorporating such measures are necessary to substantiate these hypotheses.
Reviewer Comment: The authors should expand the discussion to include more recent literature, such as this recent study published in Biomedicines 10.3390/biomedicines12112503, which highlights the emerging role of SGLT2 inhibitors and GLP-1 receptor agonists in mitigating cardiovascular risk and OSA severity through pleiotropic effects.
Response: We thank the reviewer for the suggestion. The referenced study has now been cited in the revised Discussion section (citation [23]) to broaden the context regarding pleiotropic effects.
Round 2
Reviewer 2 Report
Comments and Suggestions for Authors
My compliments to the authors for having integrated my comments in their manuscript.
Reviewer 3 Report
Comments and Suggestions for Authors
The manuscript has been appropriately revised.